# Non-Cooperation within a School-Based Wellness Program during the COVID-19 Pandemic—A Qualitative Research

**DOI:** 10.3390/ijerph19116798

**Published:** 2022-06-02

**Authors:** Moria Golan, Galia Ankori, Tamar Hager

**Affiliations:** 1Department of Nutritional Sciences, Tel-Hai College, Upper Galilee 1220800, Israel; 2Shahaf, Community Based Facility for Body Image and Eating, Ganey Hadar 7683000, Israel; 3Faculty of Social Sciences and Humanities, Tel-Hai College, Upper Galilee 1220800, Israel; ankori.galia@gmail.com (G.A.); tamar.hager@gmail.com (T.H.); 4Child and Adolescence Mental Health Clinic of Maccabi Health Services, Sderot Binyamin 21, Netanya 4231111, Israel

**Keywords:** school-based wellness program, barriers, COVID-19, preadolescents, parents’ and teachers’ engagements

## Abstract

This paper presents a qualitative analysis of COVID-19′s impact on the development, delivery, and uptake of “Favoring Myself”, a school-based interactive wellness program conducted via Zoom during 2020–2021. “Favoring Myself” targets resilience, self-esteem, body-esteem, self-care behaviors, and media literacy among 5th-grade preadolescents. Data were obtained from meetings, 23 semi-structured interviews with parents, teachers, and principals, and other modes of correspondence. All data were transcribed and thematically analyzed. The analysis highlighted the barriers faced when delivering external programs during COVID-19. Parents’ difficulties in cooperating with the program, distrustful relationships between parents and the education system, as well as teachers’ overload and stress, were identified as barriers to the external program’s sustainability. These challenges are discussed in light of previous studies of school-based programs, the psychological and social contexts of an ongoing crisis and the impact of neoliberalism on education. This study concludes that school-based prevention programs and accompanying research should be more flexible and focus on understanding and relating to parents’ and schools’ fears, uncertainties, and resistance. It is the hope of the authors that knowledge created through this exploration will be helpful in future coping vis-à-vis prevention program teams and recipients in times of unpredictable, unmanageable, and overpowering crises.

## 1. Introduction

This article explores parents’ and teachers’ engagement and difficulties in co-operating with the wellness program “Favoring Myself” and its accompanying research during the COVID-19 pandemic. “Favoring Myself”—a school-based resilience program—was integrated into the school curriculum in Israel four years ago. The program’s contents and face-to-face delivery effectiveness have been studied extensively see, for example, [1], with overall high levels of satisfaction among school principals, teachers, pupils, and their parents [2]. When the pandemic began, the first author and her team faced a dilemma–should they refrain from implementing the program as long as the pandemic persisted, or should they, to the contrary, offer their accumulated expertise in this time of crisis. Given that “Favoring Myself” provides a platform for addressing emotional, personal, and social issues, while improving resilience, coping skills, self-esteem, and body esteem, it seemed highly relevant and even crucial to the situation at hand. In addition, efforts were made to modify the program in response to the new conditions. Nevertheless, many schools were unexpectedly reluctant to participate in the program and the accompanying research and, in those that took part, harsh criticism and a general atmosphere of mistrust made it challenging to carry out the program as planned. This qualitative study describes and explores this state of affairs, hoping to shed light on the ways in which a crisis such as COVID-19 may influence ongoing external educational school programs. It is the hope of the authors that knowledge created through this exploration will be helpful in future coping by prevention program teams and recipients in times of unpredictable, unmanageable, and overpowering crises.

The following literature review explores the importance of cooperation between prevention program teams, teachers, and parents. Then, the literature on the influence of the COVID-19 pandemic on educational systems in general and mainly on remote learning is reviewed. Finally, a description of “Favoring Myself” follows, with its accompanying research, and the modifications this program has undergone to make it relevant to current conditions.

### 1.1. Literature Review

The socio-ecological scientific prevention method, as well as other universal, and multimethod programs advocate approaching multiple levels, with the understanding that promoting the strengths of children, parents, and schools may lead to multiple positive outcomes [3,4,5]. In particular, parents and teachers play important roles in raising preadolescents’ awareness of self-care behaviors and social issues [6]. Indeed, previous research has reported that shared experiences between youth and parents increased the latter’s empathy for and understanding of their children’s experiences, thus helping them to gain positive outcomes from an educational program [7]. Similarly, a motivated teacher can influence both children and parents by portraying a positive attitude towards the prevention program, ensuring its implementation [8]. Yet only a few wellness prevention programs have reported long-lasting partnerships between school teams and academics who either initiated the program or were involved in researching it [9,10]. Thus, it is vital to explore the perceptions of parents and school staff regarding content and adaptation, to identify the barriers to implementing school-based prevention programs, mainly in a state of worldwide crisis.

The recent COVID-19 outbreak has challenged youth’s ability to continue to perform tasks, develop, and achieve goals under conditions of vulnerability, helplessness, uncertainty, and a relative lack of control over outcomes [11]. During these times, boundaries have been fluid and difficult to define. The COVID-19 pandemic has created global disruption in education [12], and education systems have had to address unpredictable circumstances [13]. Most educational institutions over the world suspended face-to-face teaching and went online, and thus youth’s routines were disrupted [14,15]. Studies point to the ramifications of online studying, demonstrating, for example, that these platforms lack crucial environmental contacts [6], such as peer interactions and personal communications with teachers in the school environments [16]. In addition, the forced isolation has come at the expense of peer group attachments, and consequently, individuals’ sense of self-esteem has been hampered [15,17]. Remote learning engages only vision and hearing, blocking the ability to employ other senses mandatory to human relationships and experiences. Moreover, online platforms, with their lack of intimacy during lessons, challenge children’s ability to remain attentive and to control external cues from other technologies or people in their environment. This may damage their sense of commitment and obligation to their studies [18]. Students were no longer a “captive audience” within the educational system, and they often lacked the motivation to attend distance learning.

In terms of COVID-19′s influence on relationships, it has been found that students and their parents who are confined to their homes feel stressed and anxious and that relationships between the parents and schoolteachers become strained [19]. Both parents and school teams have felt intimidated by each other (the home environment was exposed to teachers’ critical eye and teachers’ lessons were exposed to parents’ observations and interference). School principals have reported their diminished sense of autonomy under parents’ watchful eyes [20]. Consequently, policymakers, schools, and parents have been functioning under highly challenging conditions [21]. This was the context in which “Favoring Myself” was delivered from October 2020 to January 2021. In order to better understand the program examined, the following is a more detailed description of its components.

### 1.2. The Program and Its Accompanying Research

“Favoring Myself” is an interactive, school-based, wellness program that has been integrated into the school curriculum in Israel as part of a module on life skills initiated by the National Ministry of Education in primary schools. The program highlights key protective factors to increase resilience and well-being, reduce risk behaviors and prevent internalizing unrealistic sociocultural norms. The program consists of ten 90-min weekly sessions [1,2] with each session targeting another salient topic, chosen in accordance with previous research suggesting that strengthening protective factors reduces mental health problems among adolescents [22]. The session topics and activities are detailed in Appendix A. The program is delivered by undergraduate students from the Departments of Education and Nutrition Sciences at Tel-Hai College. Program facilitators receive intensive training and supervision. Adopting the socio-ecological scientific approach of working at multiple levels [3], both parental and teacher involvements are integral components of the current phase of this program. In the four years of its implementation, this program has been constantly accompanied by an evaluative study, in which pupils have filled out questionnaires examining their resilience, self-esteem, body esteem, eating perceptions, and media literacy.

### 1.3. Pandemic Modifications

With the outbreak of COVID-19, it was clear that some modifications would have to be made in order to implement preventive strategies [23], including “Favoring Myself”:A.A unique community coordinator assisted with recruiting and addressing the difficulties caused by COVID-19.B.Duration of sessions was reduced from 90 to 60 min. Lengthy discussions were replaced by interactive activities.C.Each class was divided into two sub-groups that attended the program during different hours to enable their class teacher to participate in both.D.School-team collaboration: (1) An introductory Zoom conversation was conducted with each school team. (2) To facilitate supervision of instructors by teachers, as well as communication among all members of the program team, a computerized form was implemented. (3) Each school principal was advised to request that the head of the parents’ committee support the program.E.Engagement strategies for the 5th-grade participants included the following: (1) Two pupils were chosen in each class to assist the facilitators, motivating the group to attend and complete assignments. (2) A competition was planned between classes, measuring attendance, cooperation rate, and task submission. (3) Facilitators reached out to the fifth-grade participants before and after the program sessions, sending humorous messages as reminders to practice the skills taught in class.F.Parental component: (1) Parents received information about the program content, structure, and previous outcomes, along with a link to a short video describing the program. (2) Consenting parents received weekly updates about the topic discussed and its importance, and one or two shared assignments to be practiced with their children. (3) Two reminders were sent to participants who did not complete the weekly assignment. (4) An assignment delivered by both the child and a parent received two stars, while those produced only by the child received one star.G.Evaluative research modification: (1) A Kahoot simulation questionnaire was created to prepare pupils to answer the research questionnaire. (2) The program facilitators discussed the various feelings participants might experience when they faced questions, they were not sure of how to answer or felt embarrassed about. (3) Each subgroup of 12 to 14 participants was divided into two Zoom breakout rooms, and additional research assistants were present in each room to assist pupils who had difficulties in understanding the questionnaire.

## 2. Methodology

The current article is based on a qualitative study, aimed to understand the perceptions and opinions of parents and school teams to the implementation of “Favoring Myself” and its accompanying effectiveness research during the COVID-19 pandemic.

### 2.1. Research Approach

Choosing a qualitative approach stemmed from the search for an interpretive, subjective manner of studying people’s own perceptions of events to understand and interpret reality from their points of view, highlight and explain life experiences, and give them meaning. Qualitative research allows us to deeply explore different perspectives and to discover the complexities of situations through a holistic framework [24,25,26]. Qualitative methods are more appropriate for an understanding of reality and knowledge as constructed, rather than as ‘truth’ [27]. In addition, this method is considered appropriate for data collected in naturalistic settings, such as ours, in which the researchers are also active members of the system they study [28].

### 2.2. Participants

The four educational institutions in which the program was conducted are in the north of Israel, serving a financially comfortable population. Since data included protocols of meetings, phone calls, and WhatsApp contacts, all the people involved in the program from 2020 to 2021 are considered as participants in this study. In addition, twenty-three people agreed to be interviewed at the program’s completion. The school principals were asked to invite all the teachers and parents who were involved in the project to contact the researchers if they were willing to be interviewed regarding their experiences with “Favoring Myself”. Initially, only two school principals, three teachers, and six parents contacted the researchers. Using a snowball method, additional interviewees were recruited. Finally, two school principals, five 5th grade teachers, and sixteen parents were interviewed. The interviewed parents represented various attitudes and behavioral responses to the program that year: Six were highly engaged with the parental component of the intervention, four refused to consent to their child’s participation in the study, and six withdrew their consent for their child’s participation and involvement in the parental component.

### 2.3. Ethical Considerations

This study received approval from the College Institutional Review Board (No 12/2017/-1 and 08/2018–4, pre-registered in April 2018 under registration number NCT03540277). Parents and school teams received information about the program and the qualitative study and provided their informed consent via computerized links. All the procedures involved in this study comply with the ethical standards of the 1975 Helsinki Declaration and Consort 2010 guidelines and regulations.

### 2.4. Data Collection

Data were obtained from the program’s participants and collaborators: school principals, teachers, and parents using the following methods:
1.Data from meetings of the research team with the schools to set shared objectives, build trust, and discuss ways to address participants’ engagement.2.WhatsApp messages and telephone calls: Some parents who had concerns regarding the program sent messages to the teachers and researchers. These messages were saved. Phone calls from parents were documented by teachers, facilitators, and researchers as soon as possible after their completion.3.Zoom meeting: To avoid the escalation of parents’ agitation in one of the schools, the first author suggested that the school principal host a Zoom meeting with interested parents and school team members. A 1.5-h discussion with 30 volunteering parents and the school’s fifth-grade teachers was conducted via Zoom by the first author and the school principal, during the second week of the program. This meeting, in which parents expressed a great deal of anger and frustration, was recorded, with the consent of all participants, and then transcribed.4.Interviews: At the program conclusion, twenty-three in-depth, semi-structured personal interviews (see Appendix A for the interview guide), lasting 40–60 min, were conducted over Zoom by a research student. Interviews were audio-recorded and then transcribed.

### 2.5. Data Analysis

Data from all resources were analyzed using the thematic analysis approach. Reading and re-reading the texts was initially performed, and side notes were taken. Then, the researchers identified recurring key themes that shed light on interviewees’ experiences and attitudes. Stories of different participants which seemed to have common motives were connected, in order to represent processes [29] and understand the combined meanings of these texts [30,31].

This analysis gave us profound insight into school principals’, teachers’, and parents’ reactions toward the program as well as a deeper understanding of the relationship between the parents and the education system in times of persistent crisis.

## 3. Results

A thematic analysis of data obtained from preparatory meetings, emails, WhatsApp messages, interviews, and the Zoom meeting, identified themes, which are presented hereby divided into school reactions, parent reactions, and parent–school relationships.

### 3.1. School Reactions

The reactions in the educational systems can be divided into school’s ambivalence, teachers’ difficulties in co-operating with facilitators, and principals’ reactions.

#### 3.1.1. School Systems’ Ambivalence

Unlike in previous years, schools were reluctant to commit to the program, which manifested itself from the onset in principals’ inconsistent responsiveness to the program’s community coordinator’s calls. After receiving information letters with a link to a three-minute video highlighting the program’s importance, objectives, and novelty, only seven out of the seventeen approached schools expressed their interest. During the initial months of recruitment, principals were not ready to commit “*due to the uncertainty regarding the reopening of the schools, the teaching methods, and handling of the schools’ competing needs”*. They said, “*It is not the right time for these decisions.”* Finally, four schools chose to enroll, providing their consent only in October 2020, but only three of them adhered to the program’s agreed schedule. The fourth school slated the program time to the afternoons, which resulted in the absence of pupils who had joined a basketball class. The school’s coordinating teacher was quite upset: “*I failed to convince the principal that changing the program’s schedule would decrease participants’ motivation*”, and also noted that no one considered that changing the time of the program imposed personal challenges: *“I had to be present during these sessions after my teaching day had ended.”*

#### 3.1.2. Teachers’ Difficulty in Collaborating

Although introductory Zoom meetings were attended by the vast majority of teachers of eligible classes, and most of them expressed enthusiasm towards the program and agreed to the research demands, eventually many of them had a hard time maintaining their ongoing collaboration with the program’s facilitators. While some teachers provided helpful feedback to the group facilitators after sessions, many failed to do so consistently. In the interviews, these teachers self-confessed that they were exhausted after having experienced *“a lack of gratitude from the participants’ parents, which made it hard to remain involved and motivated”.* One of them said: *“I understand that it is an important program, and I see the facilitators’ efforts, but I cannot supervise them thoroughly. I am always in a rush to my next class.”*

In addition, only one teacher out of the four classes in which parents mounted resistance to the program, attempted to mediate and negotiate with these parents. The other teachers confessed that they *“didn’t want to fuel the fire”* and that the parents were too upset for them to *“confront them over an external program”* or that *“the parents overreacted, even violently, to the program, and to protect myself, I took a step back and maintained a cordial relationship with them.”*

The following quote from one of the teachers seems to express the complexity of their experience: “I have gone above and beyond to recruit participants to attend these lessons, stayed after my working hours. And combined with the emotional burden of my students and the lack of privacy, ‘Favoring Myself’ was not prioritized”. This teacher’s experience demonstrates how, despite good intentions, cooperation became almost impossible.

#### 3.1.3. School Principals’ Reactions

School principals varied in the way they coped with the challenges surrounding the program’s delivery during the COVID-19 pandemic.

One of the principals sent out a clarification letter to the parents, in which she explained the importance of the program as part of a mandatory lesson, and the choice of “Favoring Myself” due to its excellent reputation. Relating to parents’ concerns regarding sensitive issues she explained: *“The questionnaire and the program relate to complex and latent topics that are not discussed in most households… the child’s teacher and school counselor attend the sessions and explain the research topics to participants. Our objective is to educate a mature, aware, and resilient generation”.*

In his interview, another principal explained, “Dealing with our parents is complicated, which we know from experience. Now they are more antagonistic than ever before”. A third school principal asserted, “Despite the good intentions of the program’s facilitators and the class teacher, this year, introducing the program was an incorrect decision. It is a wonderful program with great objectives. Still, such sessions lose their effectiveness over a Zoom platform where participants’ privacy is compromised, and implementation is problematic due to the exhaustion of the system and the teachers, and the sacrifices that have been made this year.”

### 3.2. Parents’ Reactions

Parents’ reactions are divided into their reactions to the program itself and their experiences with the accompanying study.

#### 3.2.1. Parents’ Reactions to the Program

Parents’ views of the sessions varied. Some appreciated the opportunity, while others felt that *“the program includes intimidating topics that should be discussed delicately and not by young students as facilitators.”* Thus, for example, a mother praised one of the assignments but objected to uploading a picture of her and her son exercising on the class WhatsApp group: *“I do not understand why I should share our activities with others. It bothered me.”*

As mentioned earlier, one of the modifications to the program due to the pandemic was adding a competition game component in order to encourage attendance and performance of assignments by the child or the family. Competition between classes was perceived by some parents as harmful because *“if I do not like to perform the shared assignments, I am the ‘bad guy’ who causes my child’s class to receive fewer stars.”* One of the mothers stated, *“It bothered me when I heard my daughter tell her friend that if they do not complete the assignments, their group will not get the stars. This is social pressure,”* or: *“If this is a positive challenge, why bribe them?”*

Interestingly the delivery of the program over the Zoom was experienced as two-folded. On the one hand, it was perceived as problematic, as one mother stated: “*Our children sit in front of the screen for so many hours a day as it is. We may say we are in control, but the truth is we are not.”* On the other hand, and potentially as a reaction, the Zoom classes were perceived and presented by parents as an opportunity to observe teachers’ and facilitators’ conduct. One of the mothers said, *“I sat and watched many of these lessons. Some are nice, but sometimes you hear the facilitator pushing the children into a corner.”* One of the mothers stated, *“You are using our children for a damaging experiment, and I say so because this program has never been delivered on Zoom.”*

#### 3.2.2. Parents’ Reactions to the Accompanying Research

Parents’ reactions to the research were overall quite negative, varying from passive non-cooperation, such as forgetting to turn in forms they agreed to sign, to moderate complaints focusing on the burden of filling out questionnaires by their child, who already had difficulties with his/her *“existing load of assignments”,* or feeling that *“such a study should not be conducted during the COVID crisis”*, and ending with extreme reactions, such as saying that they did not want their child to be *“treated as a guineapig”* or to be *“forced to participate in the study”. “It is not acceptable that our children are obligated to answer personal questions,”* said one of the fathers.

Some parents were worried that specific themes appearing in the questionnaires would *“wake up sleeping dogs.”* For example, a mother who felt that her daughter *“never worried about her body, and as a person who suffered from eating disorders in the past, I do not want her to answer these questions.”* A somewhat more practical and moderate approach was expressed by parents who suggested that “*if the whole questionnaire and the program protocols had been sent to parents earlier, we might have felt differently.”.*

### 3.3. Parent–School Relationships

Some parents expressed anger towards the school or the school’s management. A parent wrote, “*from my point of view, clear boundaries were crossed here and under the roof of our local school.”* In relation to the study, one of the parents wondered, *“Who in our school permitted the research process?”* Other parents were more concerned with their child’s confidentiality and addressed their school’s management with questions such as “*How do you secure our children’s privacy*?” Teachers also had their own reactions to being distrusted and criticized by parents. One of the teachers commented on the parents’ protest, saying, *“The class teachers accompanied this process. You talk as if we were not there. We were there to mediate.”* A school principal described the parents as “*opinionated and suspicious”.*

## 4. Discussion

The current research addresses some of the barriers faced when attempting to establish academic–school–parent collaboration for the application of a wellness program during the COVID-19 pandemic.

According to the results, parents expressed ambivalence toward the program. Initially, they agreed to their children’s participation, and some of them were pleased with certain activities or sessions. But when they were expected to perform assignments with their children and consequently were exposed to the program’s more personal and sensitive themes, various parents reacted with anger and criticism. In response to parents’ concerns that research on body image might precipitate body or eating problems in their child, it was explained that previous research has shown a very low prevalence of distress due to children’s participation in body image research [32]. Nevertheless, although it was emphasized that preventing pupils from addressing issues such as body image and eating disorders may diminish their ability to learn something important about themselves [33], many parents were still anxious about the possibility of such harm.

Many parents were also opposed to the use of incentives to encourage attendance and performance of assignments. They talked about “bribing” and “social pressure”. General resistance was elicited, mainly when a few parents, quarantined at home, saw their children struggling to complete the research questionnaire. They saw the questionnaire as a sign of exploitation of their children by the program, the facilitators, and the researcher. Consequently, they condemned the school and program staff for damaging their children’s well-being. They were not appeased by the calming statements from the class teachers, school counselor, principal, and the first author.

Some parents were annoyed by the facilitators’ and researchers’ expectations that they would join their children’s assignments. COVID-19 has changed teaching and learning processes [34], putting what they experienced as a heavy burden on parents [35,36]. Under the volatile circumstances of social isolation, lockdowns, and online education, parents had already assumed many additional roles and wanted some autonomy regarding their extra-curricular activities with their children. Moreover, in the context of COVID-19, it seems that parents expected schools to limit novel experiences to avoid environmental stress. Previous research shows that when external facilitators offered a prevention program, even at regular times, parents were reluctant to consent to research demands [37,38]. Parents criticizing the program for being conducted by students could have been expressing a combination of refusal to handle novelty and an intensified mistrust of external educational allies.

Parents’ rageful comparisons of their children to laboratory animals, may be a reflection of worldwide disputes around the COVID-19 vaccine. Skepticism about vaccine development [39], fear of serious side effects [40] and the worry that important information regarding the vaccine may have been masked, were common in social and traditional media and may have influenced parents’ discourse around the current nonmedical study.

Their resentment and worries resemble the findings of other studies regarding the general mood during the pandemic. Current research shows that many people experienced distress, including anxiety, depression, and over-excitement [14,23]. It seems that some parental reactions could be related to the overall stress experienced worldwide at that time.

Teachers’ reactions to the program were characterized by frustration and even a sense of helplessness. This finding is consistent with Carlson’s [41] study showing that teachers experienced stress related to the uncertainty of the pandemic, the need to demonstrate flexibility, and to model caring. Educators in another study [42] expressed mixed emotions and confusion concerning their teaching lives, relating some of their stress to communication during the pandemic being primarily top-down, moving from district to principal to teacher, with themselves then bearing responsibility for communicating directly with the families. It seems that being an external program, “Favoring Myself” posed an even greater hardship on teachers who were expected to communicate with parents, while transferring messages they have not initiated regarding a program they were not teaching themselves. This non-convenient situation manifested in teachers’ explanations that given all their burdens, they could not prioritize the program, nor help the team deal with the parents *“over an external program”*.

Regarding parent–school relationships, the results resemble previous studies, which have demonstrated strained relations between parents and schoolteachers during the pandemic [19,20]. An interesting issue was that of trust. Whereas in previous years, parents expressed their faith in the program and the research team and hardly asked for information beyond what was delivered to them in the first place [1,43], this time the program elicited extreme reactions of mistrust. The parents’ suspicions may reflect their loss of trust in the education system whose functioning was erratic during the crisis. It also echoes their concern for their children’s well-being during the pandemic. It could be that they transferred their fear of possible harm to their children by the coronavirus onto the educational program. Consequently, they took decisive steps to eliminate its “threat”, thus “saving” their children from “possible harm”.

This parental response also resonates with neo-liberal education policies which promote parental educational rights, responsibilities, and involvement, while encouraging them to express their choices, and legitimizing their refusal to obey school instructions. Hornby et al. [38] have shed light on the diminished authority of local education, strengthening parents’ role as consumers. In a state of defensiveness and disintegration of the education system due to COVID-19, parents felt even more compelled to protect their children’s educational rights and diminish their frustration when confronting the program’s materials by assuming responsibility [44]. COVID-19 thus intensified the neo-liberal parenting paradox; to protect the well-being of their children, the parents saw it as their duty to reject a school-based wellness program that could have supported their children in times of crisis.

Finally, building on studies and theories supporting the effectiveness of socio-ecological prevention methods [3], “Favoring Myself” attempted to address multiple levels and create a collaborative program (schools, parents, and academics) for the benefit of children [6,7,8]. Although this seemed especially essential at the time of an overwhelming crisis, in reality, the implementation of the program evoked hard feelings in all parties, to the point that the principal who concluded that *“this year the initiation of the program was an incorrect decision*” may have articulated the experiences of the program’s team as well. Indeed, many school-based wellness interventions have been suspended during the pandemic [23].

It, therefore, seems important to suggest ways in which prevention program teams and developers may cope with future unpredictable crises. Fornaro, Struloeff, and Flowers [45] who studied external academic programs delivered during COVID-19, advocated for flexibility, exhibited as an operational adaptation while coping with uncertainty [46]. Flexibility was indeed exercised by the first author who, as mentioned, made many modifications, creatively addressing issues as they came up. Yet Fornaro, Struloeff, and Flowers [45] stress that flexibility should be an overall verbally acknowledged phenomenon manifested by and toward people in different roles in the program.

Fornaro, Struloeff, and Flowers [45] also suggest a principle they called: “Empathy for All”—the effort to hold onto an understanding that educators, families, and students face varying situations as a result of COVID-19 and therefore it is important to demonstrate caring and kindness around expectations, and attempt to value others’ time and resources. Moreover, they stress that, despite the importance of top-down decisions during crises to set policies, leaders should ensure that their staff members do not lose their own voices. Although, in “Favoring Myself”, facilitators and teachers were consulted all along, the parents may have somehow been left behind. Some of them said that they may have felt differently had they been consulted and exposed to more relevant information.

### Strengths and Limitations

This research provides essential information about the barriers faced in prevention programs during times of crisis. Twenty-three interviews, a group meeting, WhatsApp messages, and emails are regarded as a small sample, to draw general conclusions. Qualitative research, however, does not demand bigger samples since its reliability and validity are achieved through a deeper analysis of an existing sample with its multiplicity of voices [47,48]. In addition, the small sample enabled us to concentrate on the various voices and their intricate attitudes, illuminating the complexity of the situation at hand [48]. Consequently, this study sheds light on possible barriers by various stakeholders to program implementation, making an important contribution to the field of school-based prevention program research.

## 5. Conclusions

“Favoring Myself” is rooted in the socio-ecological scientific prevention approach [49] and is directed towards multiple levels of change agents [49,50], while exercising a participatory approach. This study brings to the fore the challenges facing parents and school staff collaboration with an external prevention program, in a period of uncertainty. It implies that, in certain circumstances, translating science into widespread practice is insufficient; there is a need to build processes and structures to increase trust between all parties. Future studies should investigate ways to obtain parental and school-staff cooperation beyond the consenting stage, especially where neo-liberal agendas and personal values rule education, and the organization and school management are no longer a dominant enough motivational force. In such circumstances non-collaboration of parents, teachers and external educational agencies can unintentionally limit children’s educational opportunities. When anxiety rules the educational setting, researchers should be more considerate towards the participants, perhaps even discussing with them whether to conduct the prevention program, and if so, co-constructing the venues to doing so.

## Data Availability

The data that support the findings of this study are available from the first author upon reasonable request and with permission of The College IRB. Restrictions apply to the availability of these data by the ministry of education, and so are not publicly available.

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
