# Peer review of "Non-Cooperation within a School-Based Wellness Program during the COVID-19 Pandemic—A Qualitative Research"

_ijerph, 2022, doi:10.3390/ijerph19116798_

Round 1
Reviewer 1 Report
This manuscript presents a variety of results relating to the use of the school-based prevention program ‘Favoring Myself’ during the COVID-19 pandemic. It highlights some of the challenges associated with implementing prevention programs during COVID-19. However, I found it difficult to grasp the rationale and aims of the research and also found the results difficult to follow.
Introduction
1. The literature review is somewhat limited and I am not confident that the following phrase is correct: “…only a few studies of prevention programs added parental component. Most of them have failed to recruit or retain sufficient sample sizes to allow statistical significance testing…”.
There is a huge amount of literature on the prevention of youth mental health difficulties and a reasonable number of studies have included a parental component. For large reviews see Weisz et al., 2005, Promoting and Protecting Youth Mental Health Through Evidence-Based Prevention and Treatment, American Psychologist, 60, 628-648; and Durlak et al., 2011, The impact of enhancing students’ social and emotional learning: A meta-analysis of school-based universal interventions. Child Development, 82, 405-432.
As a result of the limited literature review, it is difficult to understand why the current study is needed. Paragraph 2 of the Introduction notes “This article attempts to fill this gap” but I am not clear what the gap is. This lack of a clear rationale and clear aims for the research made it difficult to follow the rest of the manuscript.
Method
2. Design/sample: It is noted that “this is the first year that schools refused to cooperate with the randomization process…” but at this stage, no information has been provided about prior evaluations of the program. I wonder if some of the information at the start of section 2.3 could be provided earlier, to give the necessary background context to this being an ongoing program in Israel.
3. How were participants identified for the interviews? Was there an effort to seek parents who supported / did not support the program, or was this random? Did the interview sample include everyone who agreed to interviews, or could recruitment have continued if more interviews had seemed necessary? Please provide further information about how the interview sample was selected and approached.
Results
4. The results are presented in a way that doesn’t fit with the reported qualitative focus of this study. For example, under section 4.3, it is noted that 188 out of 220 parents agreed their children could participate. This isn’t information from the qualitative interviews. The study either needs to be restructured to include a quantitative element reporting on program use/uptake and a qualitative element, or otherwise these quantitative results need to come out.
It is also unclear which results have come from qualitative interviews and which come from other sources. For example, under section 4.3, were the three parents who expressed irritation towards the process of filling the questionnaire via Zoom reporting this via interview, or in another format?
Towards the end of the Results, there is then the mention of a Zoom meeting with 30 parents that gave rise to other themes. This should have been mentioned earlier with much more information about why the meeting was run and who was invited.
The whole results section is difficult to follow and hard to link back to the reported focus of the study.
Reviewer 2 Report
The topic is good and may fill the research gap. The writing contents in the Materials and Research Methodology sections are good, but there are some problems in the Results section:
It should be more elaboration on the results, especially in the parents' reaction section. As the research team did semi-structured interviews with the parents, but the results showed are lack of details and were only simply classified into 4.3,1, 4.3.2, and 4.3.3. I am willing to know the different feelings of the interviewed parents about the program instead of showing a few samples of parents annoyed with it. Do any interviewed parents feel happy and satisfied with the program? It could make as a comparison.
In 4.3, although some of the parents feel irritated after the first questionnaire. It just shows the minority (about 10%) among them. I am interested to know, do there are any comments on the other majority of parents?
In 4.3.3, there is a trend of a decline the students' participation. Does it relate to the criticism from the parents? It could be due to other reasons for the absence. If yes, it is suggested to have more elaboration on it.
In the discussions section, it could be better if deliver some suggestions to improve the program and overcome the complaints from the parents.
The abstract should not be introduced only to the research methodology but have to briefly explain the research findings.
Round 2
Reviewer 1 Report
The Authors have revised the manuscript comprehensively in response to original comments. I think it is now much clearer and a stronger study. I have just two remaining points / queries:
1. In the Abstract, the sentence "Data was obtained from meetings, semi-structured interviews with parents, teachers, and principals, and other modes of correspondence" should read "Data WERE..." - as 'data' is plural.
2. In the Method, Participants, I am still unclear how the 23 interviewees were recruited. Were additional possible participants approached for interviews but declined? On p. 6, para 2, it is noted "Reaching more interviewees would have been a challenge due to parents’ noncooperation and the small sizes of the school teams". This implies that the Authors may have liked to conduct more interviews but recruitment challenges prevented this. However, on p. 7, end of section 2.4.4., it is noted "Data saturation was judged as the point at which no new information was obtained from the interviewees". This implies saturation was reached after the 23 interviews and no more were deemed necessary. Please clarify how many people were invited to interview and how the decision was made to stop at the sample of 23.
